# Genetic Diversity and Genome-Wide Association Analysis of the Hulled/Naked Trait in a Barley Collection from Shanghai Agricultural Gene Bank

**DOI:** 10.3390/ijms25105217

**Published:** 2024-05-10

**Authors:** Zhiwei Chen, Zhenzhu Guo, Luli Li, Nigel G. Halford, Guimei Guo, Shuwei Zhang, Yingjie Zong, Shiseng Liu, Chenghong Liu, Longhua Zhou

**Affiliations:** 1Shanghai Key Laboratory of Agricultural Genetics and Breeding (21DZ2271900), Key Laboratory for Safety Assessment (Environment) of Agricultural Genetically Modified Organisms of Ministry of Agriculture and Rural Affairs (Shanghai), Shanghai Agricultural Biosafety Evaluation and Testing Professional Technical Service Platform (23DZ2290700), Biotechnology Research Institute of Shanghai Academy of Agricultural Sciences, Shanghai 201106, China; gzzsk173@163.com (Z.G.); liluli1003@163.com (L.L.); guoguimei@saas.sh.cn (G.G.); zhangshuwei@saas.sh.cn (S.Z.); zongyingjie@saas.sh.cn (Y.Z.); l406773972@outlook.com (S.L.); liuchenghong@saas.sh.cn (C.L.); 2Rothamsted Research, Harpenden AL5 2JQ, UK; nigel.halford@rothamsted.ac.uk

**Keywords:** *Hordeum vulgare* L., naked trait, genotyping-by-sequencing (GBS), genome-wide association studies (GWAS), Kompetitive Allele Specific PCR (KASP)

## Abstract

Barley is one of the most important cereal crops in the world, and its value as a food is constantly being revealed, so the research into and the use of barley germplasm are very important for global food security. Although a large number of barley germplasm samples have been collected globally, their specific genetic compositions are not well understood, and in many cases their origins are even disputed. In this study, 183 barley germplasm samples from the Shanghai Agricultural Gene Bank were genotyped using genotyping-by-sequencing (GBS) technology, SNPs were identified and their genetic parameters were estimated, principal component analysis (PCA) was preformed, and the phylogenetic tree and population structure of the samples were also analyzed. In addition, a genome-wide association study (GWAS) was carried out for the hulled/naked grain trait, and a KASP marker was developed using an associated SNP. The results showed that a total of 181,906 SNPs were identified, and these barley germplasm samples could be roughly divided into three categories according to the phylogenetic analysis, which was generally consistent with the classification of the traits of row type and hulled/naked grain. Population structure analysis showed that the whole barley population could be divided into four sub-populations (SPs), the main difference from previous classifications being that the two-rowed and the hulled genotypes were sub-divided into two SPs. The GWAS analysis of the hulled/naked trait showed that many associated loci were unrelated to the *Nud*/*nud* locus, indicating that there might be new loci controlling the trait. A KASP marker was developed for one exon-type SNP on chromosome 7. Genotyping based on the KASP assay was consistent with that based on SNPs, indicating that the gene of this locus might be associated with the hulled/naked trait. The above work not only lays a good foundation for the future utilization of this barley germplasm population but it provides new loci and candidate genes for the hulled/naked trait.

## 1. Introduction

Barley (*Hordeum vulgare* L.) is the world’s fourth largest cereal crop after wheat, rice and maize [1]. Barley is widely grown and can be used for food, feed, and brewing, with naked (free-threshing) barley being the type mainly consumed. Recently, with the strengthening of people’s health awareness, the health value of naked barley has also been recognized [2,3,4]. For example, barley is valued for its high fiber and protein content, as well as its beneficial effect on blood sugar regulation and heart health. Barley may also have anti-inflammatory and antioxidant properties, which may be beneficial for overall health and disease prevention. Therefore, the use of barley is likely to be further developed.

It is believed that the earliest domesticated barley was two-rowed and hulled because of the characteristics of wild barley (*Hordeum vulgare* L. ssp. *spontaneum*). According to archaeological evidence, barley was first domesticated between 10,500 and 9500 years ago [5]. Subsequently, six-rowed barley appeared between 8800 and 8000 years ago, and naked barley about 8000 years ago. In hulled barley, the outer parts of the flower (pales) are ‘glued’ to the grain, whereas in naked barley the pales are only loosely attached to the grain and fall off easily. Although naked barley is now found all over the world, it is common in east Asia, especially in Nepal and the Tibetan plateau. Due to the short season when crops can be grown in the highland areas, and the early ripening and stress resistance of naked barley, it has become a staple food there. We know that the hulled/naked trait is primarily controlled by a single locus (*Nud*/*nud*) that is located on chromosome arm 7 HL [6]. This locus encodes a transcription factor that regulates lipid synthesis, and thus explains the monophyletic origin of naked barley [1]. Although there have not been many reports of other genes controlling the hulled/naked trait other than *Nud*/*nud* so far, the complex biosynthesis of lipids and the enormous genetic diversity of barley suggest that naked barley in different regions may be caused by allelic variants of more than one gene. Indeed, one barley accession record from Ethiopia (HOR1143) has multiple single nucleotide polymorphisms (SNPs) associated with the hulled/naked trait on different chromosomes [7].

Barley is also one of the main cereal crops in Shanghai, China, and it has a long cultivation history in this area. A large number of barley germplasm samples have been collected and preserved, and they play an important role in barley breeding in the region. Previously, we used simple sequence repeat (SSR) markers and genotyping-by-sequencing (GBS)-based SNP markers to analyze the genetic background of 112 barley landraces, and these genotypes can be divided into three clusters according to the phylogenetic tree, which seem to be consistent with the classification according to the traits of row type and hulled/naked character [8,9]. In addition, the problem of duplicate names was also solved, based on their different genetic backgrounds, which will be more conducive to the protection and utilization of these germplasm resources. Here, we used GBS technology to analyze the other 71 barley genotypes in the Shanghai barley collection (including mainly imported varieties or genotypes, and also newly bred cultivars) to analyze the genetic diversity of all of the resources preserved in this collection. Furthermore, a genome-wide association study (GWAS) was performed to analyze the hulled/naked trait in order to identify new loci, thereby providing more evidence that multiple loci control this trait. Finally, KASP markers were developed based on SNPs associated with the hulled/naked trait, to provide a new tool for the marker-assisted selection (MAS) and speed breeding of naked barley. Meanwhile, the discovery of new loci controlling the hulled/naked trait might also provide new evidence for the origin of naked barley.

## 2. Results

### 2.1. Distribution and Classification of SNPs in the Germplasm Collection

A total of 181,906 SNPs were obtained by GBS (Appendix A). They were distributed on all seven chromosomes but were distributed more frequently at the ends of chromosomes. Chromosome 7 contained the most SNPs, with a total of 31,446 SNPs that accounted for 17.3% of the total SNPs, whereas chromosome 4 contained the lowest number of SNPs, with a total of only 17,518 SNPs, which accounted for 9.6% of the total (Figure 1A). These SNPs could be classified into three types according to their effects on protein coding: silent-type SNPs, missense-type SNPs, and nonsense-type type SNPs, which accounted for 49.96%, 49.50%, and 0.54% of the total SNPs, respectively (Figure 1B). They could also be divided into upstream-, exon-, intron-, downstream-, and intergenic-type SNPs, according to their locations in coding and non-coding DNA regions. The intergenic type of SNP was the most abundant, accounting for 58.02% of total SNPs (Figure 1B). Moreover, the SNPs located in protein-coding regions could be further classified into three types, comprising missense-type SNPs (encoding a different amino acid), nonsense-type SNPs (translation terminated) and silent-type SNPs (synonymous) (Figure 1C). The missense and silent types of SNPs were present in similar proportions, accounting between them for 99.46% of the total SNPs.

### 2.2. Genetic Diversity and Population Structure Analysis

The 181,906 SNPs used for the genetic diversity analysis were all bi-allelic (Appendix A). The number of effective alleles (Ne) ranged from 1.0202 to 2.0000, with an average of 1.4140. The main allele frequency (MAF) ranged from 0.0100 to 0.5000, with an average of 0.1800. The Hardy–Weinberg equilibrium *p*-value (HW-P) ranged from 1.16 × 10^−55^ to 1.00, with an average of 0.1130 (>0.05), indicating that this population fits the Hardy–Weinberg equilibrium. The polymorphism information content (PIC) ranged from 0.0196 to 0.3750, with an average of 0.2052 (<0.25). The expected heterozygosity (He) ranged from 0.0198 to 0.5000, with an average of 0.2517, whereas the observed heterozygosity (Ho) ranged from 0 to 1.0000, with an average of 0.0598. Nucleotide diversity (Pi) ranged from 0.0199 to 0.5017, with an average of 0.2520.

A total of 126,687 SNPs, which were further filtered by the requirement of MAF ≥ 0.05, were used for phylogenetic tree and principal component analyses. It was shown that these barley germplasm samples could be divided into three groups, which were named I, II, and III (Figure 2A). Considering the traits of row-type and the adherence of hulls to the grains, most genotypes in group I were two-rowed and hulled, except for B27, B28, and B147 (six-rowed and hulled), and excluding B115 and B158 (two-rowed and naked). In contrast, most genotypes in subgroup II were six-rowed and naked, with the exception of B006 (six-rowed and hulled), whereas most genotypes in group III were six-rowed and hulled, with the exception of B13 (two-rowed and hulled). Therefore, there was a strong relationship between molecular marker-based classification and these two morphological traits of row type and the adherence of hulls to the grains. PCA analysis results were consistent with these two morphological traits being important, with PC1 (33.65%) and PC2 (9.14%) together accounting for a total of 42.79% of the genetic variation (Figure 2B).

A total of 5848 SNPs, obtained by PLINK, were used for population structure analysis. The lowest CV was detected at K = 4, indicating that K = 4 was the best value for the population structure analysis (Figure 3A). It was indicated that there were four ancestry components for this barley germplasm collection, enabling the genotypes to be divided into four sub-populations (SPs), which were named SP1, SP2, SP3, and SP4 (Figure 3B). The genotypes in SP1 were six-rowed and hulled, except for B13 (two-rowed and hulled); in contrast, those in SP4 were two-rowed and hulled except for B27, B28 and B147 (six-rowed and hulled), B115 and B158 (two-rowed and naked), and B146 and B151 (six-rowed and naked). All of the genotypes in SP2 were six-rowed and naked, and all those in SP3 were two-rowed and hulled. Hence, three sub-populations were clearly distinguishable on the basis of row type and the adherence of hulls to the grains, with the two-rowed and hulled-barley types further divided into two sub-populations to make a total of four. Meanwhile, we also found that the three barley cultivars that had been bred by our lab were classified into different sub-populations, with Hua 30 and Hua 22 classified into SP3 and Hua11 classified into SP4.

### 2.3. Linkage Disequilibrium (LD) Analysis

LD analysis was performed using 126,687 SNPs (i) for the whole barley collection; (ii) separately for two-rowed barleys (only one accession was two-rowed and naked), six-rowed and hulled barleys, and six-rowed and naked barleys; and (iii) for each of the four sub-populations. The squared-allele frequency correlations (r2) were used for LD estimation, and they were found to decay rapidly with distance (Figure 4A). Comparing the LD of different types of barley, it was found that the decays of the two-rowed barleys and six-rowed and hulled barleys were closer and faster, whereas the decays of the six-rowed and naked barleys were slower (Figure 4B). This suggested that the level of genetic diversity in six-rowed and naked barleys was lower than that in the first two types of barleys. Similarly, for LD comparisons of different sub-populations, the decays of the SP4 group were the fastest, those of SP1 and SP3 were similar and in the middle, and SP2 showed the slowest decay (Figure 4C). Among the four sub-populations, SP4 was dominated by two-rowed and hulled barley, SP1 was dominated by six-rowed and hulled barley, SP3 was also dominated by two-rowed and hulled barely, and SP2 was dominated by six-rowed and naked barley.

### 2.4. Genome-Wide Association Study (GWAS) of the Hulled/Naked Trait

A total of 126,687 SNPs and the MLM were used for the GWAS. In this study, the quantile–quantile (Q-Q) plots showed that the MLM was suitable for association analysis (Figure 5A). The results showed that 976 and 189 significant marker–trait associations (MTAs) were detected for the hulled/naked trait based on the false discovery rate (FDR) and Bonferroni-correction methods, respectively (Figure 5B; Appendix A). Due to the large number of associated loci, subsequent analysis was performed with a higher threshold. Among the 189 MTAs, there were 2, 33, 31, 2, 17, and 104 SNPs located on 1H, 2H 3H, 4H, 6H, and 7H, respectively, with distinct peaks on chromosomes 2H, 3H, 6H, and especially 7H.

### 2.5. Associated-SNP Validation by KASP Assay across the Whole Barley Collection

A SNP (7H-523092315) located in the exon region on chromosome 7H was randomly selected for KASP marker development (Appendix A). The 150 bp upstream and downstream sequences of this SNP were used for PCR- and KASP-primer design (see Appendix A). The SNP was first confirmed by using normal PCR and Sanger sequencing, then it was used for KASP detection. After detection using the KASP marker, the whole population could be genotyped and assessed for bi-allele characteristics, indicating that the development of the KASP marker was successful (Figure 6). In general, yellow and blue dots represent the two homozygous genotypes, whereas green dots represent the heterozygous genotype. Here, there were no green dots (Figure 6), indicating that only the two homozygous genotypes existed at this locus in this whole barley collection. Comparison of the results of genotyping based on KASP and GBS analysis showed that they were completely consistent, except for B91, which was not detected in the KASP analysis, showing that the accuracy of the two techniques was very high (see Appendix A). The results of KASP and GBS genotyping with the hulled/naked trait was also completely consistent according to the chi-square test (X^2^_0.05,1_ = 3.84) (see Table 1). This suggested that this marker was closely linked to the hulled/naked trait and that the gene in which the locus was located might be a new candidate to control the trait.

## 3. Discussion

Based on the identification of naked barley germplasm samples with an intact *Nud* locus [7] and the different origins of barley domestication [10,11], the hulled/naked trait may be controlled by multiple loci or genes. Returning to the *Nud* locus itself, it is now known to encode a transcription factor from the ethylene reactive factor (ERF) family that is primarily involved in regulating lipid biosynthesis [1]. Lipid synthesis is a very complex process that involves many genes, and the deletion of some of the other genes involved may also lead to the phenotype of naked caryopses. Considering some intermediate types of the naked trait, it is also possible that the naked trait changes from a qualitative trait to a quantitative trait. Wabila et al. obtained MATs on chromosomes 2, 3, 6, and 7 by GWAS, with a false discovery rate of 5% when using the whole collection used in their study or within sub-populations [7]. Another study obtained a peak for the hulled/naked trait on chromosome 7 when using GWAS with the Bonferroni-correction method, with the most highly associated SNP located about 453 kb away from the *Nud* gene [12]. Although the role of the *Nud*/*nud* gene in controlling the hulled/naked trait has been validated by gene editing, the phenotype that was obtained was more similar to an intermediate type [13].

In this study, we identified 976 MATs in total by using the FDR method, and they were distributed across all 7 chromosomes. We further raised the threshold by using the Bonferroni-correction method, and we still obtained 189 MATs, covering all chromosomes except chromosome 5, with most MATs on chromosome 7. This means that there may be new and important loci that control the hulled/naked trait in addition to the *Nud* gene, especially on chromosome 7. Among the 104 SNPs associated with the hulled/naked trait on chromosome 7, based on the Bonferroni-correction method, nine of them are located within exons. None of the genes in which these exons are located encode an ERF family transcription factor similar to that encoded by the *Nud* locus (Appendix A). Obviously, these genes are candidate novel genes for controlling the trait. One of the associated SNPs was used to develop a KASP marker and was verified across the whole collection. Functional studies of the gene within this SNP locus may identify the novel gene controlling the hulled/naked trait and provide more evidence for the origin of naked barley.

Naked barley is called “qingke” in the Qinghai–Tibet Plateau region of China and “yuanmai” in the middle and lower reaches of the Yangtze River, and these two areas were also the main planting areas of naked barley in the past. Due to the adjustment of the planting structure, naked barley is currently mainly produced in the Qinghai–Tibet Plateau. With the continuous investigation and utilization of the nutritional and health properties of naked barley, as well as its relevance for food security, people have begun to pay attention to naked barley again in China. Therefore, the breeding of naked barley will be carried out in more areas, but due to the long-term absence of naked-barley breeding in some areas, it may be difficult to make progress. Gene editing provides a good example of how to quickly convert a better hulled-barley variety into a naked-barley variety [13]. This requires us to figure out exactly how many genes are in control of the hulled/naked trait in order to ensure that we get the ideal naked barley. As we know, there is no strict discrimination of hulled- and naked-barley varieties other than through the hulled/naked trait, so it is also feasible to breed naked-barley varieties through crosses between the two types of barley. The marker-assisted selection (MAS) technology can distinguish the individuals that appear as naked barley in the recombined and segregated offspring at the seedling stage, which greatly reduces the work required for breeding. There is no doubt that KASP marker-based detection technology has the characteristics of being high throughput and allowing visualization, and only seedling DNA is required to complete the genotyping of each sample, which is simple and efficient [14]. In this study, the SNP marker that was associated with the hulled and naked trait was obtained through GWAS and converted into KASP markers, which could serve in the MAS-based breeding of naked barley.

In addition, compared with our previous work [9], it was found that the classification based on the phylogenetic tree was similar, whereas in the population structure analysis, two sub-populations dominated by two-rowed barley varieties appeared, indicating that it might be affected by other factors in addition to morphological characters, such as geographical location and growth habits [12].

## 4. Materials and Methods

### 4.1. Plant Materials

A total of 183 barley genotypes from a Shanghai barley collection were used in this study, including 3 barley cultivars bred by our laboratory, namely Hua 30, Hua 22, and Hua 11. These barley genotypes were originally obtained from Shanghai Agricultural Biogenetic Center, and were then propagated and kept at the Shanghai Academy of Agricultural Sciences. All barley germplasm samples were grown in fields in the Huacao area of the Shanghai Academy of Agricultural Sciences (121°32′40.81″ E, 31°21′92.24″ N) during the 2021–2022 cropping season. The hulled/naked trait was directly observed from the seeds and the row type was observed in the field at the maturity stage. The information of the first 112 barley genotypes was reported previously [9], and that of the last 71 barley genotypes is listed in Table 2.

Seeds of each of the 71 barley genotypes were sown in seedling trays in an artificial-climate room at the Shanghai Academy of Agricultural Sciences. At the 3-leaf stage, leaves were collected and frozen quickly in liquid nitrogen, then stored at −80 °C for genomic DNA isolation.

### 4.2. Genotyping-by-Sequencing (GBS) and SNP Calling

Total genomic DNA isolation and sequencing-library construction for SNP arrays were performed mainly according to Li et al. [9]. GBS was conducted on the Illumina Nova platform (Illumina Inc., San Diego, CA, USA) by Oebiotech Company (Shanghai, China), and 150 bp pair-end raw reads were generated. The clean bases ranged from 1.04 to 1.62-Gb, and the clean Q30 were all >88% (see Appendix A).

The local alignment of the clean reads of the newly studied 71 barley genotypes (NCBI accession number PRJNA941018) and the previously studied 112 barley genotypes (NCBI accession number PRJNA814403) to the reference genome of barley cv. Morex (V2) was conducted using BWA software (v0.7.17) [9], and the reference genome had been previously downloaded from Ensembl Plants. SNP calling was performed using the HaplotypeCaller module of Genome Analysis Tool Kit (GATK) software (v4.1.3) and only SNP markers with QualByDepth (QD) ≥ 2 were preserved [15]. SNP markers were further filtered using vcftools (v0.1.16) and SNPs with a sequencing depth < 4, a minor allele frequency (MAF) < 0.01, or a missing rate > 20% were removed [16]. The filtered SNP markers were annotated using the SnpEff software (v4.1g) [17].

### 4.3. Genetic Diversity and Population Structure Analysis

Genetic parameters for each of filtered SNPs described above, including the Hardy–Weinberg equilibrium *p*-value (HW-P), observed heterozygosity (Ho), expected heterozygosity (He), polymorphic information content (PIC), nucleotide diversity (π), observed number of alleles (Na), and effective number of alleles (Ne) were calculated using vcftools (v0.1.16).

The SNPs used in phylogenetic tree analysis and principal component analysis (PCA) were data filtered by requiring MAF < 0.05. The phylogenetic tree of 183 barley genotypes was constructed using the neighbor-joining method [18]. The distance matrix was calculated by TreeBeST software (v1.9.2) [19], and the reliability of the tree was tested by the BootStrap method (repeated 1000 times) [20]. PCA was performed using GCTA software (v1.26.0) [21].

SNPs without tight linkage were selected for population structure analysis using PLINK (v1.9) [22]. Population structure analysis was performed by ADMIXTURE software (v1.3) with different *K* values from 2 to 10 and 10 repetitions for each *K* value [23]. The optimal *K* value for this barley population was determined by the cross validation error (CV), and the *K* value with the lowest CV was the optimal one.

### 4.4. Linkage Disequilibrium (LD) Analysis and Genome-Wide Association Studies (GWAS)

The further filtered SNPs, which were used for the phylogenetic tree and PCA analyses, were also used for LD analysis and GWAS. Linkage disequilibrium (LD) decay was estimated using PopLDdecay (v3.4) among SNP markers across the genome for the whole population [24]. Correlations between pairs of SNP markers with known chromosomal positions were used to calculate squared-allele frequency correlations (r^2^ value) [25] and the physical distance was selected to calculate LD using a fitted equation in the whole genome.

Genome-wide association scans for the hulled/naked trait and row-type trait of barley were performed using the GAPIT software (v3) with the Mixed Linear Model (MLM) [26]. Manhattan plots and quantile–quantile (Q-Q) plots were also derived using GAPIT software (v3). The threshold for significant marker–trait association was determined by both the Bonferroni-correction method and the false discovery rate (FDR) method [27,28].

### 4.5. Marker Development and Genotyping

To validate the SNPs within novel genes that were related to the hulled/naked trait, the 150 bp upstream and downstream sequences of target SNPs were used for PCR primer design. PCR primers were designed using Primer Premier software (v5.0), and DNA samples of three hulled-barley genotypes and three naked-barley genotypes were randomly selected for PCR amplifications. PCR amplifications were performed in each sample with an annealing temperature of 60 °C and a total of 35 cycles, and each reaction contained 2.5 μL 10× Buffer (within Mg^2+^), 0.5 μL of each primer (10 mM), 2.0 μL dNTP (2.5 mM), 2.0 μL rTaq enzyme (5 U/μL), and 1 μL DNA template (50 ng/μL). The PCR products were detected using a 1.5% agarose gel and then sequenced for the validation of potential SNPs.

KASP primers were designed using Primer Premier 5, and forward primers were synthesized with a FAM tag or a VIC tag at the 5 prime end of the two alleles, respectively, for each SNP. DNA samples of all barley genotypes were used for the KASP assay. The two forward primers and one common reverse primer for each SNP were mixed in a 1:1:3 ratio and used for touchdown PCR. Touchdown PCR was performed on the CFX Connect Real-Time System (BIO-RAD) with the following steps: Initial denaturation at 95 °C for 10 min, 10 touchdown cycles (95 °C for 20 s, touchdown from 61 °C to 55 °C, down 0.6 °C per cycle, each temperature for 60 s) and then 27 cycles of amplification (95 °C for 20 s, 55 °C for 60 s). Each reaction contained 2.5 mL 2× KASP master mix, 1.25 μL of primer mix and 1.25 μL of template DNA. Fluorescence data were collected and analyzed directly on the CFX Connect Real-Time System after touchdown PCR. All primers used in this study are listed in Table 3.

## 5. Conclusions

The SNPs obtained by the GPS sequencing method were used for genotyping and genetic diversity analysis in the barley collection within 183 barley genotypes, and their genetic relationships were also clarified, thus laying a foundation for better utilization of these barley germplasm resources in future. In addition, we obtained a batch of SNPs associated with the hulled/naked trait by GWAS analysis, and the genes in which these SNPs are located might be new candidates for controlling the hulled/naked trait. According to the KASP marker genotyping that was performed with the KASP marker transformed with one of the associated SNPs, the accuracy of the genotyping based on SNPs obtained from GBS and the existence of the SNP locus were both verified. These all suggested that there might be new loci that control the hulled/naked trait, providing new evidence for the origin of naked barley.

## Figures and Tables

**Figure 1 ijms-25-05217-f001:**
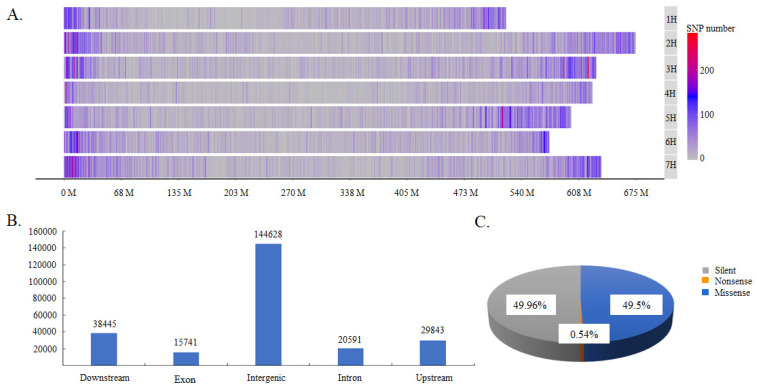
The distribution and classification of identified SNPs. (**A**). SNP distribution across chromosomes; (**B**) the classification of SNPs according to their locations in DNA regions; and (**C**) the classification of SNPs according to their effects on protein function.

**Figure 2 ijms-25-05217-f002:**
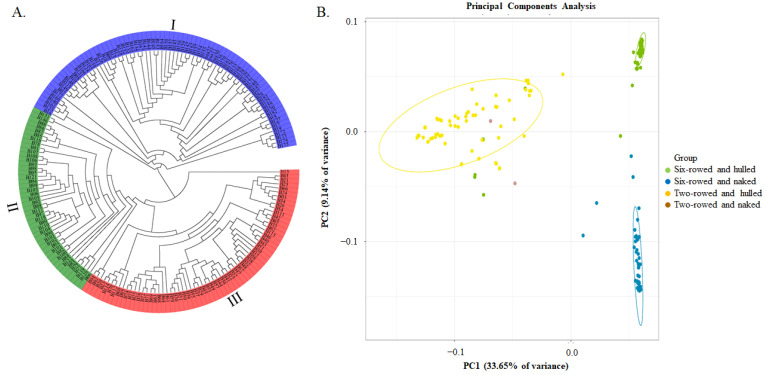
Phylogenetic tree and Principal Component Analysis (PCA) of 183 barley genotypes based on 126,687 SNP markers. (**A**) The phylogenetic tree constructed using the neighbor−joining method; and (**B**) the PCA plot, the green dots represent six-rowed and hulled barley genotypes, the blue dots represent six-rowed and naked barley genotypes, the yellow dots represent two-rowed and hulled barley genotypes, and the red dots represent six-rowed and naked barley genotypes.

**Figure 3 ijms-25-05217-f003:**
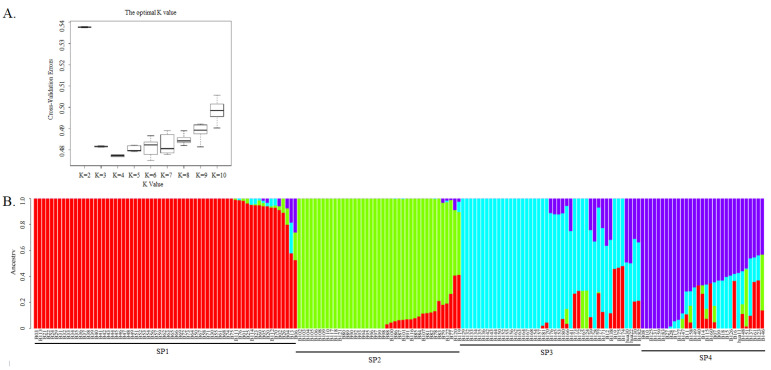
Population structure analysis of 183 barley genotypes using 5,848 SNP markers. (**A**). Estimation of possible sub-populations (SP) using the cross-validation errors (CV) at K values from 2–10; and (**B**) population structure and clustering of 183 barley genotypes with K = 4.

**Figure 4 ijms-25-05217-f004:**
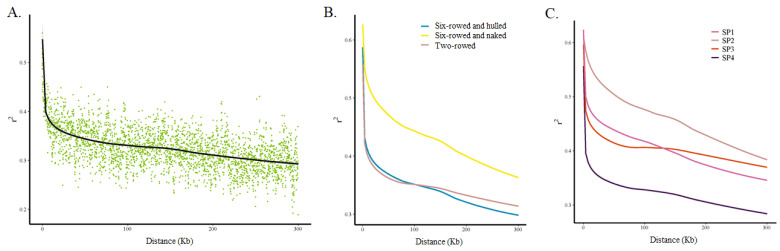
LD patterns and LD decay in the whole panel and in subgroups. (**A**). The whole panel, (**B**) subgroups according to row type and hulled/naked trait, and (**C**) subgroups according to the four sub-populations based on population structure analysis.

**Figure 5 ijms-25-05217-f005:**
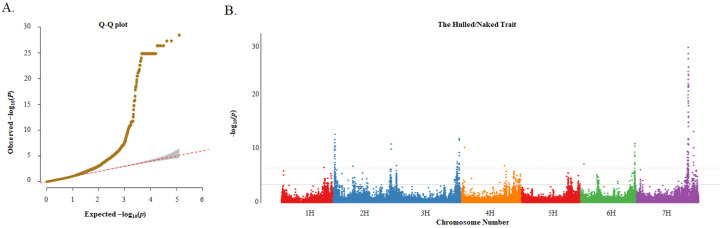
Q–Q plot and Manhattan plot for genome-wide association study (GWAS) of the hulled/naked trait in 183 barley germplasms. (**A**) Q–Q plot based on MLM model; and (**B**) Manhattan plot based on MLM model. The solid gray line and the gray dotted line indicate the significance thresholds after correction for multiple testing using the false discovery rate (FDR) and Bonferroni-correction methods, respectively.

**Figure 6 ijms-25-05217-f006:**
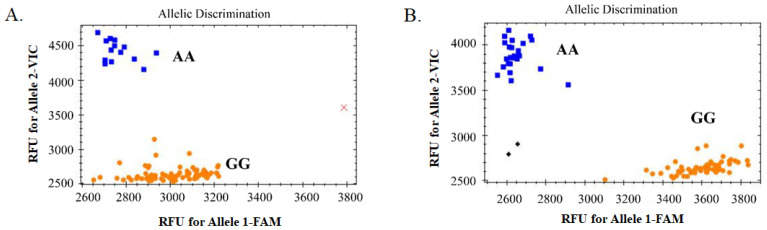
Genotype-calling screenshots of the KASP marker. (**A**). The selected 94 barley germplasm samples; and (**B**) the other 89 barley germplasm samples. The blue squares represent the homozygous genotype (AA) corresponding to the VIC-tagged primer, the orange dots represent homozygous genotype (GG) corresponding to the FAM-tagged primer, the black diamonds represent no template control (NTC), and the red cross represents the sample without fluorescence (or no detection).

**Table 1 ijms-25-05217-t001:** The chi-square test results for KASP and GBS detection methods.

	Observed	KASP	GBS
Hulled barley	138	143	143
Naked barley	45	39	40
$\chi^{2}$		0.9812	0.7367

**Table 2 ijms-25-05217-t002:** Barley germplasm samples used in this study.

Code	Name	Row Type	Adherence of Hulls	Code	Name	Row Type	Adherence of Hulls	Code	Name	Row Type	Adherence of Hulls
B113	Laotuoxu3	six	hulled	B137	Humaishisihao	two	hulled	B161	Xiaojiangdamai2	two	hulled
B114	Chiadamai	six	hulled	B138	Humaishiliuhao	two	hulled	B162	Humai4Hai2	two	hulled
B115	Mimai114	two	naked	B139	97-117	two	hulled	B163	77-130	two	hulled
B116	Ciguqing	six	naked	B140	Fengaierleng1	two	hulled	B164	76-25	two	hulled
B117	Heiliuzhu	six	naked	B141	81-18	two	hulled	B165	Jingdayihao2	two	hulled
B118	ailiuzhuyuanmai	six	naked	B142	Aizaosan	two	hulled	B166	aomaierhao	two	hulled
B119	757	six	naked	B143	Pu4	two	hulled	B167	Humaiahao2	two	hulled
B120	Aiganqi	six	naked	B144	Zhepiyihao	two	hulled	B168	Humaishihao2	two	hulled
B121	Jiadinghongjinliuzhutou2	six	naked	B145	Fuyinyihao	two	hulled	B170	Hu84-165	two	hulled
B122	Lizinyihao	six	naked	B146	Kunlunliuhao	six	naked	B171	Hu84-167	two	hulled
B123	Xiaojiangdamai1	six	hulled	B147	Mengkeer	six	hulled	B172	Hu905002	two	hulled
B124	Jingdayihao1	two	hulled	B148	78-132	two	hulled	B173	Hu905003	two	hulled
B125	Shenmaiyihao1	two	hulled	B149	Yan75-21	two	hulled	B174	Hu905004	two	hulled
B126	aoshanerleng2	two	hulled	B150	Zhou6121	two	hulled	B175	Hu905005	two	hulled
B127	Qingpuerlengdamai	two	hulled	B151	Xizhixingyeshengdamai	six	naked	B176	Hu905006	two	hulled
B128	Zhenongguangmangerleng	two	hulled	B152	85-2fu	two	hulled	B177	Hu905007	two	hulled
B129	Zaoshusanhao	two	hulled	B153	S-112	two	hulled	B180	Hu916259	two	hulled
B130	Aiaiyang	two	hulled	B154	Guangdonghuanjingua	two	hulled	B181	Hu916269	two	hulled
B131	Humaiyihao2	two	hulled	B155	Ertiaodamai	two	hulled	B182	Rudongzaosan3xuanyanxuan3	two	hulled
B132	Humaisihao1	two	hulled	B156	Zaoshuwuhao	two	hulled	B183	Shenmaiyihao2	two	hulled
B133	Humailiuhao	two	hulled	B157	Fuxuan48	two	hulled	hua30	Hua30	two	hulled
B134	Humaiahao1	two	hulled	B158	N.grinuelam	two	naked	hua22	Hua22	two	hulled
B135	Humaishihao1	two	hulled	B159	Eryuanzhufushewendingcailiaoerhao	six	hulled	hua11	Hua11	two	hulled
B136	Humaishierhao	two	hulled	B160	Fengaierleng2	two	hulled				

**Table 3 ijms-25-05217-t003:** Primers used in this study.

	Primer Name	Primer Sequence
For PCR	PCR-F	ACGCTGACGAGATGAA
	PCR-R	TCTGACTTTGGACTTGC
For KASP	KASP-F	GAAGGTGACCAAGTTCATGCTTCTGGAGCTAAGTAACCTGGCG
	KASP-V	GAAGGTCGGAGTCAACGGATTACTCTGGAGCTAAGTAACCTGGCA
	KASP-R	CACTGCTAAGGTATCTGACTTTGGACT

## Data Availability

Data are contained within the article.

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
