# Peer review of "Genetic Diversity and Genome-Wide Association Analysis of the Hulled/Naked Trait in a Barley Collection from Shanghai Agricultural Gene Bank"

_ijms, 2024, doi:10.3390/ijms25105217_

Round 1

Reviewer 1 Report

Comments and Suggestions for Authors

Dear Authors 

please find attached comments. 

Comments on the Quality of English Language

My mother tongue is not English language so I suggest to be checked.

Author Response

Dear Reviewer 1, 

Thank you very much for your review to our manuscript. We have revised the manuscript according to your comments. Please see our responses, and they are marked in red font. The revised parts in “the revised manuscript” are also in red font.

Best,

Zhiwei

Reviewer 1

The topic of this manuscript related with barley as one of the most important cereals in the world is very actual and important, specially the use of hulless barley which production is in constantly increase due to importance for human consumptions.

Abstract – should be more precise and concise, you have very interesting findings in each sections, just write it in the abstract – together with aim, material and methods and results and conclusion, on for the first read is unclear.

Revised.

Line 19 - Abstract has very scant information about aim of this manuscript, why is barley germplasm is important for global food security (you mentioned later in the introduction)

Revised.

Line 29 - please deleted that (you wrote it twice)

Deleted.

The whole abstract is not so much precise, there is no information about materials and methods, and conclusions are very unstable, try to focus on main objective, it will be important to emphasis that you analyzed genotypes using neighbor-joining method, PCA and structure analysis, GWAS

Revised.

Please, check the English through the whole text.

Checked. Our co-author Nigel G. Halford is a native English speaker.

Introduction

Line 50 – please erase „It is reasonable to believe“ try to write in a scientific way, with some concrete evidence and citation, do not use words from everyday life.

Revised.

Line 78 – this sentences (material, GWAS analysis and KASP) are related with Material and Methods section

This work is a continuation of the previous one, and it is mainly emphasized here that 71 of the 183 barley germplasm resources were sequenced in this study.

Introduction is too short, you do not explain why do you use GBS markers for further analysis, add some information from other authors work on this topic.

Please check line 71 to 78 in the first version, and this was also explained in detail in reference 9.

Line 89 – change „but to be densier“ with distributed more at the ends of chromosomes

Changed.

Line 89 - change „7th chromosome“ with chromosome 7

Changed.

Line 90 - change „4th chromosome“ with chromosome 4

Changed.

Results

are clear presented, with minor corrections with spaces in few lines and sentences where you use >, < to harmonize in the text.

Line 208 – delete space after VIC

Deleted.

Discussion

Line 214 - delete „it is reasonable to believe“. The hulled 7naked trait may be controlled by multiple loci and genes, which is results of GWAS analysis, where you observed a lot of associations on almost all chromosomes. It could be false positive association, because that trait is simple trait.

Deleted. And the simple trait may also be caused by complex physiological and biochemical reactions, and this may be the reason why there were so many associations. Other researchers also observed similar results, such as Wabila et al.

Line 221 - Wabila et al Should be changed with the number in the reference.

There is the number of the reference at the end of the sentence.

Line 238 - how do you choose that certain SNP for development a KASP, what was Your criteria?

We do this randomly, but only if this SNP is real and that no other SNPs are present within 150 bp of each side of it, or if it does not affect the primer design. Of course, we first consider SNPs located in the exons.

Line 266 - why do you not include the information about growth type habits to check– you did not have it or some other reason?

We have seen the difference in population structure analysis, and I think there may be an influence on the growth habits, while we mainly focused on the naked/hulled trait, so we only included a few traits that were obvious and important in evolution.

  1. Materials and Methods

How do you choose those additional 71 genotypes –what was the criteria? The problem is that you already used that material for publishing and you add significantly less number of genotypes than in previous study?! The size of the plot? The rest of molecular analysis are very well explained, with mentioned all software

We used all the barley germplasm in the collection, and I know that these barley germplasm were collected or bred by barley researchers and breeders in Shanghai Academy of Agricultural Sciences for decades and preserved in the gene bank. These 71 germplasm were sequenced in this study, but when analyzing, we put together the previously sequenced barley germplasm (that is 112 plus 71).

Line 344 - Initial could be in a small letter

Do you mean the PCR? It's an abbreviation.

Line 356 – what is the next step?

Revised.

Reviewer 2 Report

Comments and Suggestions for Authors

The main question addressed by this manuscript is the genetic characterization of barley germplasm from the Shanghai Agricultural Gene Bank, utilizing genotyping-by-sequencing technology. The manuscript aims to investigate the genetic relationships among these germplasms and conduct a genome-wide association study for the hulled/naked grain trait.

After reading the manuscript contributes original insights into the genetic composition and relationships of barley germplasm, which is important for global food security. By utilizing genotyping-by-sequencing technology, the research offers a novel approach to understanding the genetic diversity of barley germplasm, addressing a notable gap in the field. Also, identifying new loci and candidate genes associated with the hulled/naked grain trait adds further relevance and originality to the manuscript.

Compared to generally accessible literature resources, this manuscript stands out for its comprehensive analysis of barley germplasm utilizing genotyping-by-sequencing technology and its focus on the hulled/naked grain trait. While previous studies have explored the genetic diversity of barley germplasm to some extent, this manuscript provides a more detailed and nuanced understanding, mainly through its genome-wide association study analysis. Identifying new loci associated with the hulled/naked grain trait represents a significant advancement in the field, distinguishing this study from existing literature and highlighting its importance for future research and agricultural practices.

The Introduction is a relatively short but informative and readable chapter. At the end of the chapter, please add the aim of the study and working hypotheses.

The Material and Methods section is listed after the discussion, which is not ideal, but it is probably possible considering the rules of MDPI journals. This section contains all the essentials. The only thing that needs to be added is information about the statistical processing of the data, what methods the authors used, and what software they worked on - please add it.

Results: Besides clear graphs and tables, additional materials are added as appendices. The results are well done and short, too.

Discussion - in terms of content, this section is informative and somewhat summarizing. From a professional point of view, it is okay. However, I recommend that the findings be discussed again with other published works on a similar topic because the results are discussed with a relatively low number of authors (less than 10). Therefore, something could have been overlooked.

The conclusions are consistent with the evidence and arguments presented. In the end, however, I still need a specific summary of where the research should go next about the authors' results - please add it.

Comments on the Quality of English Language

The English level is good, and the text effectively communicates the study's findings. 

The sentences are clear and concise, and the information is presented logically. 

However, there are a few minor spelling and grammar issues.

Author Response

Dear Reviewer 2, 

Thank you very much for your review to our manuscript. We have revised the manuscript according to your comments. Please see our responses, and they are marked in red font. The revised parts in “the revised manuscript” are also in red font.

Best,

Zhiwei

Reviewer 2

The main question addressed by this manuscript is the genetic characterization of barley germplasm from the Shanghai Agricultural Gene Bank, utilizing genotyping-by-sequencing technology. The manuscript aims to investigate the genetic relationships among these germplasms and conduct a genome-wide association study for the hulled/naked grain trait.

After reading the manuscript contributes original insights into the genetic composition and relationships of barley germplasm, which is important for global food security. By utilizing genotyping-by-sequencing technology, the research offers a novel approach to understanding the genetic diversity of barley germplasm, addressing a notable gap in the field. Also, identifying new loci and candidate genes associated with the hulled/naked grain trait adds further relevance and originality to the manuscript.

Compared to generally accessible literature resources, this manuscript stands out for its comprehensive analysis of barley germplasm utilizing genotyping-by-sequencing technology and its focus on the hulled/naked grain trait. While previous studies have explored the genetic diversity of barley germplasm to some extent, this manuscript provides a more detailed and nuanced understanding, mainly through its genome-wide association study analysis. Identifying new loci associated with the hulled/naked grain trait represents a significant advancement in the field, distinguishing this study from existing literature and highlighting its importance for future research and agricultural practices.

Thank you very much.

The Introduction is a relatively short but informative and readable chapter. At the end of the chapter, please add the aim of the study and working hypotheses.

Added.

The Material and Methods section is listed after the discussion, which is not ideal, but it is probably possible considering the rules of MDPI journals. This section contains all the essentials. The only thing that needs to be added is information about the statistical processing of the data, what methods the authors used, and what software they worked on - please add it.

Please check the Material and Methods part.

Results: Besides clear graphs and tables, additional materials are added as appendices. The results are well done and short, too.

Discussion - in terms of content, this section is informative and somewhat summarizing. From a professional point of view, it is okay. However, I recommend that the findings be discussed again with other published works on a similar topic because the results are discussed with a relatively low number of authors (less than 10). Therefore, something could have been overlooked.

There is not much work in this area, and we have included most of them.

The conclusions are consistent with the evidence and arguments presented. In the end, however, I still need a specific summary of where the research should go next about the authors' results - please add it.

Added.

Reviewer 3 Report

Comments and Suggestions for Authors

Dear Authors of the manuscript "Genetic Diversity and Genome-Wide Association Analysis of the Hulled/Naked Trait in a Barley Collection from Shanghai Agricultural Gene Bank", I have the impression that the proposed manuscript is scientifically relevant and appropriately presented.

I am free to suggest a minor addition to the manuscript. Specifically, the study investigates "Genetic Diversity" and "Genome-wide association" in a barley collection from the Shanghai Agricultural Gene Bank. I think it would be interesting to provide a brief overview or recommendations on how such analyses can assist in more effectively collecting plant culture collections and improving the management of the Agricultural Gene Bank. This is not mandatory, but I think it would be of interest to readers.

Author Response

Dear Reviewer 3, 

Thank you very much for your review to our manuscript. We have revised the manuscript according to your comments. Please see our responses, and they are marked in red font. The revised parts in “the revised manuscript” are also in red font.

Best,

Zhiwei

Reviewer 3

Dear Authors of the manuscript "Genetic Diversity and Genome-Wide Association Analysis of the Hulled/Naked Trait in a Barley Collection from Shanghai Agricultural Gene Bank", I have the impression that the proposed manuscript is scientifically relevant and appropriately presented.

Thank you very much.

I am free to suggest a minor addition to the manuscript. Specifically, the study investigates "Genetic Diversity" and "Genome-wide association" in a barley collection from the Shanghai Agricultural Gene Bank. I think it would be interesting to provide a brief overview or recommendations on how such analyses can assist in more effectively collecting plant culture collections and improving the management of the Agricultural Gene Bank. This is not mandatory, but I think it would be of interest to readers.

The importance of crop germplasm resources is well known, which is why various crop germplasm resources are being collected in many parts of the world. Compared with the collection and preservation of crop germplasm resources, the evaluation and utilization of these germplasm resources are more important. This barley germplasm resource collection not only preserves the efforts of Shanghai barley breeders for decades, but also provides a proof of Shanghai barley researchers for the world's barley research. This study not only analyzed the relationship between these barley germplasm resources at the molecular level, but also provided an example of how to use these resources, highlighting the importance of these germplasm resources. In the previous study on low nitrogen tolerance of barley (Novel low-nitrogen stress-responsive long non-coding RNAs (lncRNA) in barley landrace B968 (Liuzhutouzidamai) at seedling stage. BMC Plant Biology 2020, 20, 142.), the barley germplasm was also derived from this barley germplasm resource collection.